# GreaseLM: Graph REASoning Enhanced Language Models for Question Answering

**Xikun Zhang, Antoine Bosselut, Michihiro Yasunaga, Hongyu Ren**
**Percy Liang, Christopher D. Manning, Jure Leskovec**
Stanford University
{xikunz2,antoineb,myasu,hyren,pliang,manning,jure}@cs.stanford.edu

## Abstract

Answering complex questions about textual narratives requires reasoning over both stated context and the world knowledge that underlies it. However, pretrained language models (LM), the foundation of most modern QA systems, do not robustly represent latent relationships between concepts, which is necessary for reasoning. While knowledge graphs (KG) are often used to augment LMs with structured representations of world knowledge, it remains an open question how to effectively fuse and reason over the KG representations and the language context, which provides situational constraints and nuances. In this work, we propose GreaseLM, a new model that fuses encoded representations from pretrained LMs and graph neural networks over multiple layers of modality interaction operations. Information from both modalities propagates to the other, allowing language context representations to be grounded by structured world knowledge, and allowing linguistic nuances (*e.g.*, negation, hedging) in the context to inform the graph representations of knowledge. Our results on three benchmarks in the commonsense reasoning (*i.e.*, CommonsenseQA, OpenbookQA) and medical question answering (*i.e.*, MedQA-USMLE) domains demonstrate that GreaseLM can more reliably answer questions that require reasoning over both situational constraints and structured knowledge, even outperforming models $8\times$ larger.[1]

## 1 Introduction

Question answering is a challenging task that requires complex reasoning over both explicit constraints described in the textual context of the question, as well as unstated, relevant knowledge about the world (*i.e.*, knowledge about the domain of interest). Recently, large pretrained language models fine-tuned on QA datasets have become the dominant paradigm in NLP for question answering tasks (Khashabi et al., 2020). After pretraining on an extreme-scale collection of general text corpora, these language models learn to implicitly encode broad knowledge about the world, which they are able to leverage when fine-tuned on a domain-specific downstream QA task. However, despite the strong performance of this two-stage learning procedure on common benchmarks, these models struggle when given examples that are distributionally different from examples seen during fine-tuning (McCoy et al., 2019). Their learned behavior often relies on simple (at times spurious) patterns to offer shortcuts to an answer, rather than robust, structured reasoning that effectively fuses the explicit information provided by the context and implicit external knowledge (Marcus, 2018).

On the other hand, massive knowledge graphs (KG), such as Freebase (Bollacker et al., 2008), Wikidata (Vrandečić & Krötzsch, 2014), ConceptNet (Speer et al., 2017), and Yago (Suchanek et al., 2007) capture such external knowledge explicitly using triplets that capture relationships between entities. Previous research has demonstrated the significant role KGs can play in structured reasoning and query answering (Ren et al., 2020; 2021; Ren & Leskovec, 2020). However, extending these reasoning advantages to general QA (where questions and answers are expressed in natural language and not easily mapped to strict logical queries) requires finding the right integration of knowledge from the KG with the information and constraints provided by the QA example. Prior

---

[1]All code, data and pretrained models are available at `https://github.com/snap-stanford/GreaseLM`.

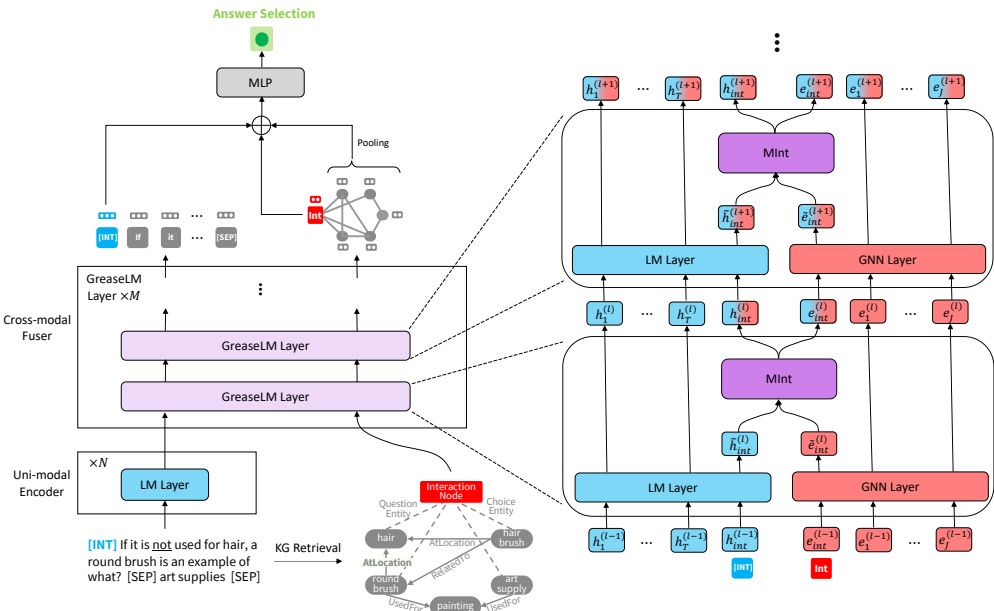

Figure 1: **GREASELM Architecture**. The textual context is appended with a special *interaction token* and passed through $N$ LM-based unimodal encoding layers. Simultaneously, a local KG of relevant knowledge is extracted and connected to an *interaction node*. In the later GREASELM layers, the language representation continues to be updated through LM layers and the KG is processed using a GNN, simulating reasoning over its knowledge. In each layer, after each modality's representation is updated, the representations of the *interaction token* and *node* are pulled, concatenated, and passed through a modality interaction (MInt) unit to mix their representations. In subsequent layers, the mixed information from the *interaction* elements mixes with their respective modalities, allowing knowledge from the KG to affect the representations of individual tokens, and context from language to affect fine-grained entity knowledge representations in the GNN.

methods propose various ways to leverage both modalities (*i.e.*, expressive large language models and structured KGs) for improved reasoning (Mihaylov & Frank, 2018; Lin et al., 2019; Feng et al., 2020). However, these methods typically fuse the two modalities in a shallow and non-interactive manner, encoding both separately and fusing them at the output for a prediction, or using one to augment the input of the other. Consequently, previous methods demonstrate restricted capacity to exchange useful information between the two modalities. It remains an open question how to effectively fuse the KG and LM representations in a truly unified manner, where the two representations can interact in a non-shallow way to simulate structured, situational reasoning.

In this work, we present GREASELM, a new model that enables fusion and exchange of information from both the LM and KG in multiple layers of its architecture (see Figure 1). Our proposed GREASELM consists of an LM that takes as input the natural language context, as well as a graph neural network (GNN) that reasons over the KG. After each layer of the LM and GNN, we design an interactive scheme to bidirectionally transfer the information from each modality to the other through specially initialized interaction representations (*i.e.*, *interaction* token for the LM; *interaction* node for the GNN). In such a way, all the tokens in the language context receive information from the KG entities through the interaction token and the KG entities indirectly interact with the tokens through the interaction node. By such a deep integration across all layers, GREASELM enables joint reasoning over both the language context and the KG entities under a unified framework agnostic to the specific language model or graph neural network, so that both modalities can be contextualized by the other.

GREASELM demonstrates significant performance gains across different LM architectures. We perform experiments on several standard QA benchmarks: CommonsenseQA, OpenbookQA and MedQA-USMLE, which require external knowledge across different domains (commonsense reasoning and medical reasoning) and use different KGs (ConceptNet and Disease Database). Across both domains, GREASELM outperforms comparably-sized prior QA models, including strong fine-

tuned LM baselines (by 5.5%, 6.6%, and 1.3%, respectively) and state-of-the-art KG+LM models (by 0.9%, 1.8%, and 0.5%, respectively) on the three competitive benchmarks. Furthermore, with the deep fusion of both modalities, GREASELM exhibits strong performance over baselines on questions that exhibit textual nuance, such as resolving multiple constraints, negation, and hedges, and which require effective reasoning over both language context and KG.

## 2 RELATED WORK

Integrating KG information has become a popular research area for improving neural QA systems. Some works explore using two-tower models to answer questions, where a graph representation of knowledge and language representation are fused with no interaction between them (Wang et al., 2019). Other works seek to use one modality to ground the other, such as using an encoded representation of a linked KG to augment the textual representation of a QA example (*e.g.*, Knowledgeable Reader, Mihaylov & Frank, 2018; KagNet, Lin et al., 2019; KT-NET, Yang et al., 2019). Others reverse the flow of information and use a representation of the text (*e.g.*, final layer of LM) to provide an augmentation to a graph reasoning model over an extracted KG for the example (*e.g.*, MHGRN, Feng et al., 2020; Lv et al., 2020). In all of these settings, however, the interaction between both modalities is limited as information between them only flows one way.

More recent approaches explore deeper integrations of both modalities. Certain approaches learn to access implicit knowledge encoded in LMs (Bosselut et al., 2019; Petroni et al., 2019; Hwang et al., 2021) by training on structured KG data, and then use the LM to generate local KGs that can be used for QA (Wang et al., 2020; Bosselut et al., 2021). However, these approaches discard the static KG once they train the LM on its facts, losing important structure that can guide reasoning. More recently, QA-GNN (Yasunaga et al., 2021) proposed to jointly update the LM and GNN representations via message passing. However, they use a single pooled representation of the LM to seed the textual component of this joint structure, limiting the updates that can be made to the textual representation. In contrast to prior works, we propose to make individual token representations in the LM and node representations in the GNN mix for multiple layers, enabling representations of both modalities to reflect particularities of the other (e.g., knowledge grounds language; language nuances specifies which knowledge is important). Simultaneously, we retain the individual structure of both modalities, which we demonstrate improves QA performance substantially (§5).

Additionally, some works explore integrating knowledge graphs with language models in the pretraining stage. However, much like for QA, the modality interaction is typically limited to knowledge feeding language (Zhang et al., 2019; Shen et al., 2020; Yu et al., 2020), rather than designing interactions across multiple layers. Sun et al. (2020)'s work is perhaps most similar, but they do not use the same interaction bottleneck, requiring high-precision entity mention spans for linking, and they limit expressivity through shared modality parameters for the LM and KG.

## 3 PROPOSED APPROACH: GREASELM

In this work, we augment large-scale language models (Devlin et al., 2019; Liu et al., 2019; Lan et al., 2020; Liu et al., 2021) with graph reasoning modules over KGs. Our method, GREASELM (depicted in Figure 1), consists of two stacked components: (1) a set of unimodal LM layers which learn an initial representation of the input tokens, and (2) a set of upper cross-modal GREASELM layers which learn to jointly represent the language sequence and linked knowledge graph, allowing textual representations formed from the underlying LM layers and a graph representation of the KG to mix with one another. We denote the number of LM layers as $N$, and the number of GREASELM layers as $M$. The total number of layers in our model is $N + M$.

**Notation.** In the task of *multiple choice question answering (MCQA)*, a generic MCQA-type dataset consists of examples with a context paragraph $c$, a question $q$ and a candidate answer set $\mathcal{A}$, all expressed in text. In this work, we also assume access to an external knowledge graph (KG) $\mathcal{G}$ that provides background knowledge that is relevant to the content of the multiple choice questions.

Given a QA example $(c, q, \mathcal{A})$, and the KG $\mathcal{G}$ as input, our goal is to identify which answer $a \in \mathcal{A}$ is correct. Without loss of generality, when an operation is applied to an arbitrary answer, we refer to that answer as $a$. We denote a sequence of tokens in natural language as $\{w_1, \ldots, w_T\}$, where $T$ is

the total number of tokens, and the representation of a token $w_t$ from the $\ell$-th layer of the model as $\boldsymbol{h}_t^{(\ell)}$. We denote a set of nodes from the KG as $\{e_1, \ldots, e_J\}$, where $J$ is the total number of nodes, and the representation of a node $e_j$ in the $\ell$-th layer of the model as $\boldsymbol{e}_j^{(\ell)}$.

## 3.1 INPUT REPRESENTATION

We concatenate our context paragraph $c$, question $q$, and candidate answer $a$ with separator tokens to get our model input $[c; q; a]$ and tokenize the combined sequence into $\{w_1, \ldots, w_T\}$. Second, we use the input sequence to retrieve a subgraph of the KG $\mathcal{G}$ (denoted $\mathcal{G}_{\text{sub}}$), which provides knowledge from the KG that is relevant to this QA example. We denote the set of nodes in $\mathcal{G}_{\text{sub}}$ as $\{e_1, \ldots, e_J\}$.

**KG Retrieval.** Given each QA context, we follow the procedure from Yasunaga et al. (2021) to retrieve the subgraph $\mathcal{G}_{\text{sub}}$ from $\mathcal{G}$. We describe this procedure in Appendix B.1. Each node in $\mathcal{G}_{\text{sub}}$ is assigned a type based on whether its corresponding entity was linked from the context $c$, question $q$, answer $a$, or as a neighbor to these nodes. In the rest of the paper, we use "KG" to refer to $\mathcal{G}_{\text{sub}}$.

**Interaction Bottlenecks.** In the cross-modal GREASELM layers, information is fused between both modalities, for which we define a special *interaction token* $w_{int}$ and a special *interaction node* $e_{int}$ whose representations serve as the bottlenecks through which the two modalities interact (§3.3). We prepend $w_{int}$ to the token sequence and connect $e_{int}$ to all the *linked* nodes $\mathcal{V}_{\text{linked}}$ in $\mathcal{G}_{\text{sub}}$.

## 3.2 LANGUAGE PRE-ENCODING

In the unimodal encoding component, given the sequence of tokens $\{w_{int}, w_1, \ldots, w_T\}$, we first sum the token, segment, and positional embeddings for each token to compute its $\ell=0$ input representation $\{\boldsymbol{h}_{int}^{(0)}, \boldsymbol{h}_1^{(0)}, \ldots, \boldsymbol{h}_T^{(0)}\}$, and then compute an output representation for each layer $\ell$:

$$\{\boldsymbol{h}_{int}^{(\ell)}, \boldsymbol{h}_1^{(\ell)}, \ldots, \boldsymbol{h}_T^{(\ell)}\} = \text{LM-Layer}(\{\boldsymbol{h}_{int}^{(\ell-1)}, \boldsymbol{h}_1^{(\ell-1)}, \ldots, \boldsymbol{h}_T^{(\ell-1)}\}) \tag{1}$$
$$\text{for } \ell = 1, \ldots, N$$

where LM-Layer$(\cdot)$ is a single LM encoder layer, whose parameters are initialized using a pretrained model (§4.1). We refer readers to Vaswani et al. (2017) for technical details of these layers.

## 3.3 GREASELM

GREASELM uses a cross-modal fusion component to inject information from the KG into language representations and information from language into KG representations. The GREASELM layer is designed to separately encode information from both modalities, and fuse their representations using the bottleneck of the special interaction token and node. It is comprised of three components: (1) a transformer LM encoder block which continues to encode the language context, (2) a GNN layer that reasons over KG entities and relations, and (3) a modality interaction layer that takes the unimodal representations of the interaction token and interaction node and exchanges information through them. We discuss these three components below.

**Language Representation.** In the $\ell$-th GREASELM layer, the input token embeddings $\{\boldsymbol{h}_{int}^{(N+\ell-1)}, \boldsymbol{h}_1^{(N+\ell-1)}, \ldots, \boldsymbol{h}_T^{(N+\ell-1)}\}$ are fed into additional transformer LM encoder blocks that continue to encode the textual context based on the LM's pretrained representations:

$$\{\tilde{\boldsymbol{h}}_{int}^{(N+\ell)}, \tilde{\boldsymbol{h}}_1^{(N+\ell)}, \ldots, \tilde{\boldsymbol{h}}_T^{(N+\ell)}\} = \text{LM-Layer}(\{\boldsymbol{h}_{int}^{(N+\ell-1)}, \boldsymbol{h}_1^{(N+\ell-1)}, \ldots, \boldsymbol{h}_T^{(N+\ell-1)}\}) \tag{2}$$
$$\text{for } \ell = 1, \ldots, M$$

where $\tilde{\boldsymbol{h}}$ corresponds to pre-fused embeddings of the language modality. As we will discuss below, because $\boldsymbol{h}_{int}^{N+\ell-1}$ will encode information received from the knowledge graph representation, these late language encoding layers will also allow the token representations to mix with KG knowledge.

**Graph Representation.** The GREASELM layers also encode a representation of the local KG $\mathcal{G}_{\text{sub}}$ linked from the QA example. To represent the graph, we first compute initial node embeddings $\{\boldsymbol{e}_1^{(0)}, \ldots, \boldsymbol{e}_J^{(0)}\}$ for the retrieved entities using pretrained KG embeddings for these nodes (§4.1). The initial embedding of the interaction node $\boldsymbol{e}_{int}^0$ is initialized randomly.

Then, in each layer of the GNN, the current representation of the node embeddings $\{e_{int}^{(\ell-1)}, e_1^{(\ell-1)}, \ldots, e_J^{(\ell-1)}\}$ is fed into the layer to perform a round of information propagation between nodes in the graph and yield pre-fused node embeddings for each entity:

$$\{\tilde{e}_{int}^{(\ell)}, \tilde{e}_1^{(\ell)}, \ldots, \tilde{e}_J^{(\ell)}\} = \text{GNN}(\{e_{int}^{(\ell-1)}, e_1^{(\ell-1)}, \ldots, e_J^{(\ell-1)}\}) \tag{3}$$
$$\text{for } \ell = 1, \ldots, M$$

where GNN corresponds to a variant of graph attention networks (Veličković et al., 2018) that is a simplification of the method of Yasunaga et al. (2021). The GNN computes node representations $\tilde{e}_j^{(\ell)}$ for each node $e_j \in \{e_1, \ldots, e_J\}$ via message passing between neighbors on the graph.

$$\tilde{e}_j^{(\ell)} = f_n\left( \sum_{e_s \in \mathcal{N}_{e_j} \cup \{e_j\}} \alpha_{sj} m_{sj} \right) + e_j^{(\ell-1)} \tag{4}$$

where $\mathcal{N}_{e_j}$ represents the neighborhood of an arbitrary node $e_j$, $m_{sj}$ denotes the message one of its neighbors $e_s$ passes to $e_j$, $\alpha_{sj}$ is an attention weight that scales the message $m_{sj}$, and $f_n$ is a 2-layer MLP. The messages $m_{sj}$ between nodes allow entity information from a node to affect the model's representation of its neighbors, and are computed in the following manner:

$$r_{sj} = f_r(\tilde{r}_{sj}, u_s, u_j) \tag{5} \qquad m_{sj} = f_m(e_s^{(\ell-1)}, u_s, r_{sj}) \tag{6}$$

where $u_s, u_j$ are node type embeddings, $\tilde{r}_{sj}$ is a relation embedding for the relation connecting $e_s$ and $e_j$, $f_r$ is a 2-layer MLP, and $f_m$ is a linear transformation. The attention weights $\alpha_{sj}$ scale the contribution of each neighbor's message by its importance, and are computed as follows:

$$q_s = f_q(e_s^{(\ell-1)}, u_s) \tag{7} \qquad k_j = f_k(e_j^{(\ell-1)}, u_j, r_{sj}) \tag{8}$$

$$\gamma_{sj} = \frac{q_s^\top k_j}{\sqrt{D}} \tag{9} \qquad \alpha_{sj} = \frac{\exp(\gamma_{sj})}{\sum_{e_s \in \mathcal{N}_{e_j} \cup \{e_j\}} \exp(\gamma_{sj})} \tag{10}$$

where $f_q$ and $f_k$ are linear transformations and $u_s, u_j, r_{sj}$ are defined the same as above.

As discussed in the following paragraph, message passing between the interaction node $e_{int}$ and the nodes from the retrieved subgraph will allow information from text that $e_{int}$ receives from $w_{int}$ to propagate to the other nodes in the graph.

**Modality Interaction.** Finally, after using a transformer LM layer and a GNN layer to update token embeddings and node embeddings respectively, we use a *modality interaction layer* (MInt) to let the two modalities *fuse* information through the bottleneck of the interaction token $w_{int}$ and the interaction node $e_{int}$. We concatenate the pre-fused embeddings of the interaction token $\tilde{h}_{int}^{(i)}$ and interaction node $\tilde{e}_{int}^{(i)}$, pass the joint representation through a mixing operation (MInt), and then split the output post-fused embeddings into $h_{int}^{(i)}$ and $e_{int}^{(i)}$:

$$[h_{int}^{(\ell)}; e_{int}^{(\ell)}] = \text{MInt}([\tilde{h}_{int}^{(\ell)}; \tilde{e}_{int}^{(\ell)}]), \tag{11}$$

We use a two-layer MLP as our MInt operation, though other fusion operators could be used to mix the representation. All the tokens other than the interaction token $w_{int}$ and all the nodes other than the interaction node $e_{int}$ are not involved in the modality interaction process: $w^{(\ell)} = \tilde{w}^{(\ell)}$ for $w \in \{w_1, \ldots, w_T\}$ and $e^{(\ell)} = \tilde{e}^{(\ell)}$ for $e \in \{e_1, \ldots, e_J\}$. However, they receive information from the interaction representations $h_{int}^{(\ell)}$ and $e_{int}^{(\ell)}$ in the next layers of their respective modal propagation (*i.e.*, Eqs. 2, 3). Consequently, across multiple GREASELM layers, information propagates between both modalities (see Fig. 1 for visual depiction), grounding language representations to KG knowledge, and knowledge representations to contextual constraints.

**Learning & Inference.** For the MCQA task, given a question $q$ and an answer $a$ from all the candidates $\mathcal{A}$, we compute the probability of $a$ being the correct answer as $p(a \mid q, c) \propto \exp(\text{MLP}(h_{int}^{(N+M)}, e_{int}^{(M)}, g))$, where $g$ denotes attention-based pooling of $\{e_j^{(M)} \mid e_j \in \{e_1, \ldots, e_J\}\}$ using $h_{int}^{(N+M)}$ as a query. We optimize the whole model end-to-end using the cross entropy loss. At inference time, we predict the most plausible answer as $\arg\max_{a \in \mathcal{A}} p(a \mid q, c)$.

| Dataset | Example |
|---|---|
| CommonsenseQA | A weasel has a thin body and short legs to easier burrow after prey in a what? (A) tree (B) mulberry bush (C) chicken coop (D) viking ship **(E) rabbit warren** |
| OpenbookQA | Which of these would let the most heat travel through? (A) a new pair of jeans **(B) a steel spoon in a cafeteria** (C) a cotton candy at a store  (D) a calvin klein cotton hat |
| MedQA-USMLE | A 57-year-old man presents to his primary care physician with a 2-month history of right upper and lower extremity weakness. He noticed the weakness when he started falling far more frequently while running errands. Since then, he has had increasing difficulty with walking and lifting objects. His past medical history is significant only for well-controlled hypertension, but he says that some members of his family have had musculoskeletal problems. His right upper extremity shows forearm atrophy and depressed reflexes while his right lower extremity is hypertonic with a positive Babinski sign. Which of the following is most likely associated with the cause of this patients symptoms? (A) HLA-B8 haplotype        (B) HLA-DR2 haplotype **(C) Mutation in SOD1**        (D) Mutation in SMN1 |

Table 1: Examples of the MCQA task for each of the datasets evaluated in this work.

## 4 EXPERIMENTAL SETUP

We evaluate GREASELM on three diverse multiple-choice question answering datasets across two domains: *CommonsenseQA* (Talmor et al., 2019) and *OpenBookQA* (Mihaylov et al., 2018) as commonsense reasoning benchmarks, and *MedQA-USMLE* (Jin et al., 2021) as a clinical QA task.

**CommonsenseQA** is a 5-way multiple-choice question answering dataset of 12,102 questions that require background commonsense knowledge beyond surface language understanding. We perform our experiments using the in-house data split of Lin et al. (2019) to compare to baseline methods.

**OpenbookQA** is a 4-way multiple-choice question answering dataset that tests elementary scientific knowledge. It contains 5,957 questions along with an open book of scientific facts. We use the official data splits from Mihaylov & Frank (2018).

**MedQA-USMLE** is a 4-way multiple-choice question answering dataset, which requires biomedical and clinical knowledge. The questions are originally from practice tests for the United States Medical License Exams (USMLE). The dataset contains 12,723 questions. We use the original data splits from Jin et al. (2021).

### 4.1 IMPLEMENTATION & TRAINING DETAILS

**Language Models.**    We seed GREASELM with RoBERTa-Large (Liu et al., 2019) for our experiments on CommonsenseQA, AristoRoBERTa (Clark et al., 2019) for our experiments on OpenbookQA, and SapBERT (Liu et al., 2021) for our experiments on MedQA-USMLE, demonstrating GREASELM's generality with respect to language model initializations. Hyperparameters for training these models can be found in Appendix Table 7.

**Knowledge Graphs.** We use *ConceptNet* (Speer et al., 2017), a general-domain knowledge graph, as our external knowledge source $\mathcal{G}$ for both CommonsenseQA and OpenbookQA. It has 799,273 nodes and 2,487,810 edges in total. For MedQA-USMLE, we use a self-constructed knowledge graph that integrates the Disease Database portion of the Unified Medical Language System (UMLS; Bodenreider, 2004) and DrugBank (Wishart et al., 2018). The knowledge graph contains 9,958 nodes and 44,561 edges. Additional information about node initialization and hyperparameters for preprocessing these KGs can be found in Appendix B.2.

### 4.2 BASELINE METHODS

**Fine-tuned LMs.**    To study the effect of using KGs as external knowledge sources, we compare our method with vanilla fine-tuned LMs, which are knowledge-agnostic. We fine-tune RoBERTa-

Table 2: **Performance comparison on *CommonsenseQA* in-house split** (controlled experiments). As the official test is hidden, here we report the in-house Dev (IHdev) and Test (IHtest) accuracy, following the data split of Lin et al. (2019). Experiments are controlled using same seed LM.

| Methods | IHdev-Acc. (%) | IHtest-Acc. (%) |
|---|---|---|
| RoBERTa-Large (w/o KG) | 73.1 ($\pm$0.5) | 68.7 ($\pm$0.6) |
| RGCN (Schlichtkrull et al., 2018) | 72.7 ($\pm$0.2) | 68.4 ($\pm$0.7) |
| GconAttn (Wang et al., 2019) | 72.6 ($\pm$0.4) | 68.6 ($\pm$1.0) |
| KagNet (Lin et al., 2019) | 73.5 ($\pm$0.2) | 69.0 ($\pm$0.8) |
| RN (Santoro et al., 2017) | 74.6 ($\pm$0.9) | 69.1 ($\pm$0.2) |
| MHGRN (Feng et al., 2020) | 74.5 ($\pm$0.1) | 71.1 ($\pm$0.8) |
| QA-GNN (Yasunaga et al., 2021) | 76.5 ($\pm$0.2) | 73.4 ($\pm$0.9) |
| GREASELM (**Ours**) | **78.5** ($\pm$0.5) | **74.2** ($\pm$0.4) |

Table 3: **Test Accuracy comparison on *OpenBookQA*.** Experiments are controlled using the same seed LM for all LM+KG methods.

| Model | Acc. |
|---|---|
| AristoRoBERTa (no KG) | 78.4 |
| + RGCN | 74.6 |
| + GconAttn | 71.8 |
| + RN | 75.4 |
| + MHGRN | 80.6 |
| + QA-GNN | 82.8 |
| GREASELM (**Ours**) | **84.8** |

Table 4: **Test accuracy comparison to public *OpenBookQA* model implementations.** *UnifiedQA (11B params) and T5 (3B) are 30x and 8x larger than our model.

| Model | Acc. | # Params |
|---|---|---|
| ALBERT (Lan et al., 2020) + KB | 81.0 | ~235M |
| HGN (Yan et al., 2020) | 81.4 | ≥355M |
| AMR-SG (Xu et al., 2021) | 81.6 | ~361M |
| ALBERT + KPG (Wang et al., 2020) | 81.8 | ≥235M |
| QA-GNN (Yasunaga et al., 2021) | 82.8 | ~360M |
| T5* (Raffel et al., 2020) | 83.2 | **~3B** |
| T5 + KB (Pirtoaca) | 85.4 | **≥11B** |
| UnifiedQA* (Khashabi et al., 2020) | **87.2** | **~11B** |
| GREASELM (**Ours**) | 84.8 | ~359M |

Large (Liu et al., 2019) for *CommonsenseQA*, and AristoRoBERTa[2] (Clark et al., 2019) for *OpenbookQA*. For *MedQA-USMLE*, we use a state-of-the-art biomedical language model, SapBERT (Liu et al., 2021), which is an augmentation of PubmedBERT (Gu et al., 2022) that is trained with entity disambiguation objectives to allow the model to better understand entity knowledge.

**LM+KG models.** We also evaluate GREASELM's ability to exploit its knowledge graph augmentation by comparing with existing LM+KG methods: (1) Relation Network (RN; Santoro et al., 2017), (2) RGCN (Schlichtkrull et al., 2018), (3) GconAttn (Wang et al., 2019), (4) KagNet (Lin et al., 2019), (5) MHGRN (Feng et al., 2020), and (6) QA-GNN (Yasunaga et al., 2021). QA-GNN is the existing top-performing model under this LM+KG paradigm. The key difference between GREASELM and these baseline methods is that they do not fuse the representations of both modalities across multiple interaction layers, allowing the representation of both modalities to affect the other (§3.3). For fair comparison, we use the same LM to initialize these baselines as for our model.

## 5 EXPERIMENTAL RESULTS

Our results in Tables 2 and 3 demonstrate a consistent improvement on the *CommonsenseQA* and *OpenbookQA* datasets. On *CommonsenseQA*, our model's test performance improves by 5.5% over fine-tuned LMs and 0.9% over existing LM+KG models. On *OpenbookQA*, these improvements are magnified, with 6.4% over raw LMs, and 2.0% over the prior best LM+KG system, QA-GNN. The boost over QA-GNN suggests that GREASELM's multi-layer fusion component that passes information between the text and KG representations is more expressive than LM+KG methods which do

---

[2]*OpenbookQA* provides an extra corpus of scientific facts in a textual form. AristoRoBERTa is based off RoBERTa-Large, but uses the facts corresponding to each question, prepared by Clark et al. (2019), as an additional input along with the QA context.

Table 5: Performance of GREASELM on the *CommonsenseQA* IH-dev set on complex questions with semantic nuance such as prepositional phrases, negation terms, and hedge terms.

| Model | # Prepositional Phrases | | | | | Negation Term | Hedge Term |
|---|---|---|---|---|---|---|---|
| | 0 | 1 | 2 | 3 | 4 | | |
| $n$ | 210 | 429 | 316 | 171 | 59 | 83 | 167 |
| RoBERTa-Large | 66.7 | 72.3 | 76.3 | 74.3 | 69.5 | 63.8 | 70.7 |
| QA-GNN | **76.7** | 76.2 | 79.1 | 74.9 | 81.4 | 66.2 | 76.0 |
| GREASELM (**Ours**) | 75.7 | **79.3** | **80.4** | **77.2** | **84.7** | **69.9** | **78.4** |

not integrate such sustained interaction between both modalities. We also achieve competitive results to other systems on the leaderboard of *OpenbookQA* (Table 4), posting the third highest score. However, we note that the T5 (Raffel et al., 2020) and UnifiedQA (Khashabi et al., 2020) models are pretrained models with $8\times$ and $30\times$ more parameters, respectively, than our model. Among models with comparable parameter counts, GREASELM achieves the highest score. An ablation study on different model components and hyperparameters is reported in Appendix C.1.

**Quantitative Analysis.** Given these overall performance improvements, we investigated whether GREASELM's improvements were reflected in questions that required more complex reasoning. Because we had no gold structures from these datasets to categorize the reasoning complexity of different questions, we defined three proxies: the number of prepositional phrases in the questions, the presence of negation terms, and the presence of hedging terms. We use the number of prepositional phrases as a proxy for the number of explicit reasoning constraints being set in the questions. For example, the *CommonsenseQA* question in Table 1, "A weasel has a thin body and short legs to easier burrow after prey in a what?" has three prepositional phrases: *to easier burrow*, *after prey*, *in a what*, which each provide an additional search constraint for the answer (*n.b.*, in certain cases, the prepositional phrases do not provide constraints that are needed for selecting the correct answer). The presence of negation and hedging terms stratifies our evaluation to questions that have explicit negation mentions (*e.g.*, *no*, *never*) and terms indicating uncertainty (*e.g.*, *sometimes*; *maybe*).

Our results in Table 5 demonstrate that GREASELM generally outperforms RoBERTa-Large and QA-GNN for both questions with negation terms and hedge terms, indicating GREASELM handles contexts with nuanced constraints. Furthermore, we also note that GREASELM performs better than the baselines across all questions with prepositional phrases, our measure for reasoning complexity. QA-GNN and GREASELM perform comparably on questions with no prepositional phrases, but the increasing complexity of questions requires deeper cross-modal fusion between language and knowledge representations. While QA-GNN's end fusion approach of initializing a node in the GNN from the LM's final representation of the context is an effective approach, it compresses the language context to a single vector before allowing interaction with the KG, potentially limiting the cross-relationships between language and knowledge that can be captured (see example in Figure 2). Interestingly, we note that both GREASELM and QA-GNN significantly outperform RoBERTa-Large even when no prepositional phrases are in the question. We hypothesize that some of these questions may require less reasoning, but require specific commonsense knowledge that RoBERTa may not have learned during pretraining (*e.g.*, "What is a person considered a bully known for?").

**Qualitative Analysis.** In Figure 2, we examine GREASELM's node-to-node attention weights induced by the GNN layers of the model, and analyze whether they reflect more expressive reasoning steps compared to QA-GNN. Figure 2 shows an example from the CommonsenseQA IH-dev set. In this example, GREASELM correctly predicts that the answer is "airplane" while QA-GNN makes an incorrect prediction, "motor vehicle". For both models, we perform Best First Search (BFS) on the retrieved KG subgraph $\mathcal{G}_{sub}$ to trace high attention weights from the interaction node (purple).

For GREASELM, we observe that the attention by the interaction node increases on the "bug" entity in the intermediate GNN layers, but drops again by the final layer, resembling a suitable intuition surrounding the hedge term "unlikely". Meanwhile, the attention on "windshield" consistently increases across all layers. For QA-GNN, the attention on "bug" increases over multiple layers. As "bug" is mentioned multiple times in the context, it may be well-represented in QA-GNN's context node initialization, which is never reformulated by language representations, unlike in GREASELM.

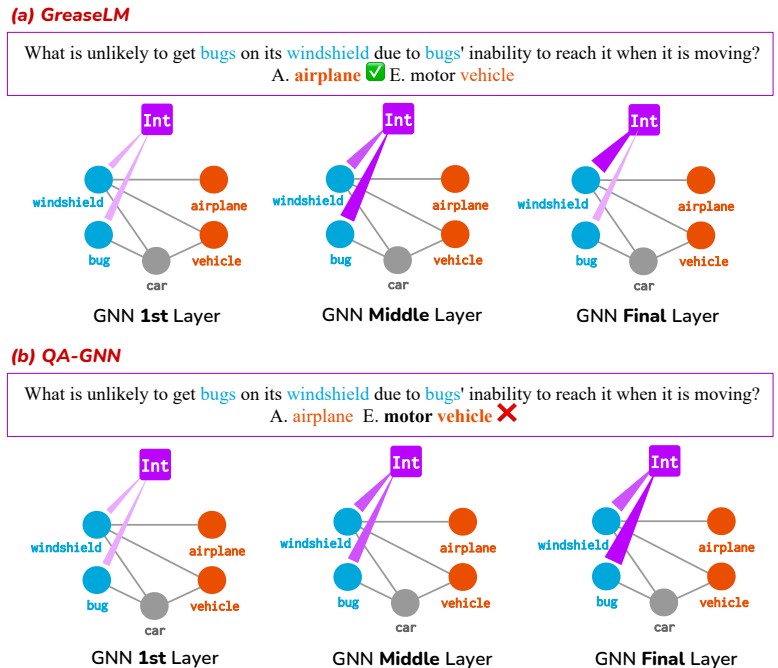

Figure 2: Qualitative analysis of GREASELM's graph attention weight changes across multiple layers of message passing compared with QA-GNN. GREASELM demonstrates attention change patterns that more closely resemble the expected change in focus on the "bug" entity.

**Domain generality** Our reported results thus far demonstrate the viability of our method in the general commonsense reasoning domain. In this section, we explore whether GREASELM could be adapted to other domains by evaluating on the *MedQA-USMLE* dataset. Our results in Table 6 demonstrate that GREASELM outperforms state-of-the-art fine-tuned LMs (e.g., SapBERT; Liu et al., 2021) and a QA-GNN augmentation of SapBERT. Additionally, we note the improved performance over all classical methods and LM methods first reported in Jin et al. (2021). Additional results in Appendix C show that our approach is also agnostic to the language model used with improvements recorded by GREASELM when it is seeded with other LMs, such as PubmedBERT (Gu et al., 2022), and BioBERT (Lee et al.,

Table 6: Performance on *MedQA-USMLE*

| Methods | Acc. (%) |
|---|---|
| **Baselines** (Jin et al., 2021) | |
| CHANCE | 25.0 |
| PMI | 31.1 |
| IR-ES | 35.5 |
| IR-CUSTOM | 36.1 |
| CLINICALBERT-BASE | 32.4 |
| BIOROBERTA-BASE | 36.1 |
| BIOBERT-BASE | 34.1 |
| BIOBERT-LARGE | 36.7 |
| **Baselines** (Our implementation) | |
| SapBERT-Base (w/o KG) | 37.2 |
| QA-GNN | 38.0 |
| GREASELM (**Ours**) | **38.5** |

2020). While these results are promising as they suggest that GREASELM is an effective augmentation of pretrained LMs for different domains and KGs (i.e., the medical domain with the DDB + Drugbank KG), there is still ample room for improvement on this task.

## 6 CONCLUSION

In this paper, we introduce GREASELM, a new model that enables interactive fusion through joint information exchange between knowledge from language models and knowledge graphs. Experimental results demonstrate superior performance compared to prior KG+LM and LM-only baselines across standard datasets from multiple domains (commonsense and medical). Our analysis shows improved capability modeling questions exhibiting textual nuances, such as negation and hedging.

ACKNOWLEDGMENT

We thank Rok Sosic, Maria Brbic, Jordan Troutman, Rajas Bansal, and our anonymous reviewers for discussions and for providing feedback on our manuscript. We thank Xiaomeng Jin for help with data preprocessing. We also gratefully acknowledge the support of DARPA under Nos. HR00112190039 (TAMI), N660011924033 (MCS); ARO under Nos. W911NF-16-1-0342 (MURI), W911NF-16-1-0171 (DURIP); NSF under Nos. OAC-1835598 (CINES), OAC-1934578 (HDR), CCF-1918940 (Expeditions), IIS-2030477 (RAPID), NIH under No. R56LM013365; Stanford Data Science Initiative, Wu Tsai Neurosciences Institute, Chan Zuckerberg Biohub, Amazon, JPMorgan Chase, Docomo, Hitachi, Intel, JD.com, KDDI, Toshiba, NEC, and UnitedHealth Group. J. L. is a Chan Zuckerberg Biohub investigator. The content is solely the responsibility of the authors and does not necessarily represent the official views of the funding entities.

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

# A  ETHICS STATEMENT

We outline potential ethical issues with our work below. First, GREASELM is a method to fuse language representations and knowledge graph representations for effective reasoning about textual situations. Consequently, GREASELM could reflect many of the same biases and toxic behaviors exhibited by language models and knowledge graphs that are used to initialize it. For example, prior large-scale language models have been shown to encode biases about race, gender, and other demographic attributes (Sheng et al., 2020). Because GREASELM is seeded with pretrained language models that often learn these patterns, it is possible to reflect them in open-world settings. Second, the ConceptNet knowledge graph (Speer et al., 2017) used in this work has been shown to encode stereotypes (Mehrabi et al., 2021), rather than completely clean commonsense knowledge. If GREASELM were used outside these standard benchmarks in conjunction with ConceptNet as a KG, it might rely on unethical relationships in its knowledge resource to arrive at conclusions. Consequently, while GREASELM could be used for applications outside these standard benchmarks, we would encourage implementers to use the same precautions they would apply to other language models and methods that use noisy knowledge sources.

Another source of ethical concern is the use of the MedQA-USMLE evaluation. While we find clinical reasoning using language models and knowledge graphs to be an interesting testbed for GREASELM and for joint language and reasoning models in general, we do not encourage users to use these models for real world clinical prediction, particularly at these performance levels.

# B  EXPERIMENTAL SETUP DETAILS

## B.1  ENTITY LINKING

Given each QA context, we follow the procedure from Yasunaga et al. (2021) to retrieve the subgraph $\mathcal{G}_{\text{sub}}$ from $\mathcal{G}$. First, we perform entity linking to $\mathcal{G}$ to retrieve an initial set of nodes $\mathcal{V}_{\text{linked}}$. Second, we add any bridge entities that are in a 2-hop path between any pair of linked entities in $\mathcal{V}_{\text{linked}}$ to get the set of retrieved entities $\mathcal{V}_{\text{retrieved}}$. Then we prune the set of nodes $\mathcal{V}_{\text{retrieved}}$ using a relevance score computed for each node. To compute the relevance score, we follow the procedure of Yasunaga et al. (2021) – we concatenate the node name with the context of the QA example, and pass it through a pre-trained LM, using the output score of the node name as the relevance score. We only retain the top 200 scores nodes and prune the remaining ones. Finally, we retrieve all the edges that connect any two nodes in $\mathcal{V}_{\text{sub}}$, forming the retrieved subgraph $\mathcal{G}_{\text{sub}}$. Each node in $\mathcal{G}_{\text{sub}}$ is assigned a type according to whether its corresponding entity was linked from the context $c$, question $q$, answer $a$, or from a bridge path.

## B.2  GRAPH INITIALIZATION

To compute initial node embeddings (§3.3) for entities retrieved in $\mathcal{G}_{\text{sub}}$ from ConceptNet, we follow the method of MHGRN (Feng et al., 2020). We convert knowledge triples in the KG into sentences using pre-defined templates for each relation. Then, these sentences are fed into a BERT-large LM to compute embeddings for each sentence. Finally, for all sentences containing an entity, we extract all token representations of the entity's mention spans in these sentences, mean pool over these representations and project this mean-pooled representation.

For MedQA-USMLE, node embeddings are initialized similarly using the pooled token output embeddings of the entity name from the SapBERT model (described in §4.2; Liu et al., 2021). For MedQA, 5% of examples do not yield a retrieved entity. In these cases, we represent the graph using a dummy node initialized with 0. In essence, GreaseLM backs off to only using LM representations as the graph propagates no information.

## B.3  HYPERPARAMETERS

Table 7: Hyperparameter settings for models and experiments

| Category | Hyperparameter | Dataset | | |
|---|---|---|---|---|
| | | CommonsenseQA | OpenbookQA | MedQA-USMLE |
| Model architecture | Number of GREASELM layers $M$ | 5 | 6 | 3 |
| | Number of Unimodal LM layers $N$ | 19 | 18 | 9 |
| | Number of attention heads in GNN | 2 | 2 | 2 |
| | Dimension of node embeddings and the messages in GNN | 200 | 200 | 200 |
| | Dimension of MLP hidden layers (except MInt operator) | 200 | 200 | 200 |
| | Number of hidden layers of MLPs | 1 | 1 | 1 |
| | Dimension of MInt operator hidden layer | 400 | 200 | 400 |
| Regularization | Dropout rate of the embedding layer, GNN layers and fully-connected layers | 0.2 | 0.2 | 0.2 |
| Optimization | Learning rate of parameters in LM | 1.00E-05 | 1.00E-05 | 5.00E-05 |
| | Learning rate of parameters not in LM | 1.00E-03 | 1.00E-03 | 1.00E-03 |
| | Number of epochs in which LM's parameters are kept frozen | 4 | 4 | 0 |
| | Optimizer | RAdam | RAdam | RAdam |
| | Learning rate schedule | constant | constant | constant |
| | Batch size | 128 | 128 | 128 |
| | Number of epochs | 30 | 70 | 20 |
| | Max gradient norm (gradient clipping) | 1.0 | 1.0 | 1.0 |
| Data | Max number of nodes | 200 | 200 | 200 |
| | Max number of tokens | 100 | 100 | 512 |

# C  ADDITIONAL EXPERIMENTAL RESULTS

## C.1  ABLATION STUDIES

In Table 8, we summarize an ablation study conducted using the CommonsenseQA IHdev set.

**Modality interaction.** A key component of GREASELM is the connection of the LM to the GNN via the modality interaction module (Eq. 11). If we remove modality interaction, the performance drops significantly, from 78.5% to 76.5% (approximately the performance of QA-GNN). Integrating the modality interaction in every other layer instead of consecutive layers also hurts performance. A possible explanation is that skipping layers could impede learning consistent representations across layers for both the LM and the GNN, a property which may be desirable given we initialize the model using a pretrained LM's weights (*e.g.*, RoBERTa). We also find that sharing parameters between modality interaction layers (Eq. 11) outperforms not sharing, possibly because our datasets are not very large (*e.g.*, 10k for CommonsenseQA), and sharing parameters helps prevent overfitting.

Table 8: **Ablation study** of our model components, using the CommonsenseQA IH-dev set.

| Ablation Type | Ablation | Dev Acc. |
|---|---|---|
| GREASELM | - | 78.5 |
| Modality Interaction | No interaction | 76.5 |
| | Interaction in every other layer | 76.3 |
| Interaction Layer Parameter Sharing | No parameter sharing | 77.1 |
| Number of GREASELM layers ($M$) | $M = 4$ | 77.7 |
| | $M = 6$ | 78.0 |
| | $M = 7$ | 76.2 |
| Graph Connectivity | Interaction node connected to all nodes in $\mathcal{V}_{sub}$, not only $\mathcal{V}_{linked}$ | 77.6 |
| Node Initialization | Random | 60.8 |
| | TransE (Bordes et al., 2013) | 77.7 |

**Number of GREASELM layers.** We find that $M = 5$ GREASELM layers achieves the highest performance. However, both the results for $M = 4$ and $M = 6$ are relatively close to the top performance, indicating our method is not overly sensitive to this hyperparameter.

**Graph connectivity.** The interaction node $e_{int}$ is a key component of GREASELM that bridges the interaction between the KG and the text. Selecting which nodes in the KG are directly connected to $e_{int}$ affects the rate at which information from different portions of the KG can reach the text representations. We find that connecting $e_{int}$ KG nodes explicitly linked to the input text performs best. Connecting $e_{int}$ to all nodes in the subgraph (*e.g.*, bridge entities) hurts performance (-0.9%), possibly because the interaction node is overloaded by having to attend to all nodes in the graph (up to 200). By connecting the interaction node only to linked entities, each linked entity serves as a filter for relevant information that reaches the interaction node.

**KG node embedding initialization.** Effectively initializing KG node representations is critical. When we initialize nodes randomly instead of using the BERT-based initialization method from Feng et al. (2020), the performance drops significantly (78.5%→60.8%). While using standard KG embeddings (*e.g.*, TransE; Bordes et al., 2013) recovers much of the performance drop (77.7%), we still find that using BERT-based entity embeddings performs best.

## C.2 EFFECT OF LM INITIALIZATION ON GREASELM

Table 9: Performance on the in-house splits of *CommonsenseQA* for different LM initializations of our method, GREASELM.

| Methods | IHdev-Acc. | IHtest-Acc. |
|---|---|---|
| ROBERTA-LARGE | 73.1 | 68.7 |
| + GREASELM (**Ours**) | **78.5** | **74.2** |
| ROBERTA-BASE | 65.1 | 59.8 |
| + GREASELM (**Ours**) | **69.3** | **65.0** |

Table 10: Initialization on *MedQA-USMLE*

| Methods | Acc. (%) |
|---|---|
| SAPBERT-BASE | 37.2 |
| + GREASELM (**Ours**) | **38.5** |
| BIOBERT-BASE | 34.1 |
| + GREASELM (**Ours**) | **34.6** |
| PUBMEDBERT-BASE | 38.0 |
| + GREASELM (**Ours**) | **38.7** |

To evaluate whether our method is agnostic to the LM used to seed the GreaseLM layers, we replace the LMs we use in previous experiments (RoBERTa-large for CommonsenseQA and SapBERT for MedQA-USMLE) with RoBERTa-base for CommonsenseQA, and BioBERT and PubmedBERT for MedQA-USMLE. Across multiple LM initializations in two domains, our results demonstrate that GREASELM can provide a consistent improvement for multiple LMs when used as a modality junction between KGs and language.

