# OpenReview forum: "GreaseLM: Graph REASoning Enhanced Language Models"
_ICLR.cc/2022/Conference — ICLR 2022 Spotlight_

### Official Review · Reviewer_RNQN · 2021-10-29

**Correctness:** 4
**Technical Novelty And Significance:** 2
**Empirical Novelty And Significance:** 3
**Recommendation:** 6
**Confidence:** 4

**Main Review:**

Strengths:
1. The idea of multi-layer fusion of two modality through additional interaction token and node is interesting.
2. The model achieves good results on three multi-choice question answering datasets.
3. The ablation study shows some interesting finds such as the importance weight sharing in Mint layers.

Weakness:
1. The technical contribution is somewhat small: except the multi-layer fusion part, others are very similar to QA-GNN including the GNN model.
2. As the main contribution is about the modality interaction layer, more analysis should be made such as the effects of number of fusion layers $M$, why only perform interaction through one single embedding (interaction token and node) instead of directly applying cross attention through all the tokens and nodes? There are also some other ways to perform embedding interaction such as the one in ERNIE [1] Figure 2 (b). To answer these questions, both intuitive analysis and empirical results are needed.

Additional Reviews:
1. Since this paper only focuses on question answering tasks, I suggest adding "for question answering" into the title similar to previous works QA-GNN and MHGRN to be more accurate.
2. There are also some related but missing references such as GLM[1], JAKET[2], CoLAKE[3]. Although they focus on language model pre-training, the idea of combining graph reasoning and language modeling is related.

[1] Zhang Z, Han X, Liu Z, et al. ERNIE: Enhanced language representation with informative entities[J]. arXiv preprint arXiv:1905.07129, 2019.
[2] Shen T, Mao Y, He P, et al. Exploiting structured knowledge in text via graph-guided representation learning[J]. arXiv preprint arXiv:2004.14224, 2020.
[3] Yu D, Zhu C, Yang Y, et al. Jaket: Joint pre-training of knowledge graph and language understanding[J]. arXiv preprint arXiv:2010.00796, 2020.
[4] Sun T, Shao Y, Qiu X, et al. Colake: Contextualized language and knowledge embedding[J]. arXiv preprint arXiv:2010.00309, 2020.

**Summary Of The Paper:**

This paper proposes a model which fuses representations from language models and graph neural networks through multiple interactions layers for multi-choice question answering tasks, which outperforms baseline methods over three datasets from two different domains.

**Summary Of The Review:**

In summary, this paper proposes an interesting idea to combine language model and graph neural networks for question answering and shows good results. But more experiments and analysis should be conducted to justify their design on modality interaction modeling. Adding them will make it a strong paper.

---

> ### Author Response · Authors · 2021-11-19
> **Response to Reviewer RNQN**
>
> We thank the reviewer for viewing our method and results as interesting. We respond to their concerns below (and in the general response above):
>
> > As the main contribution is aboeut the modality interaction layer, more analysis should be made such as the effects of number of fusion layers, why only perform interaction through one single embedding (interaction token and node) instead of directly applying cross attention through all the tokens and nodes? There are also some other ways to perform embedding interaction such as the one in ERNIE [1] Figure 2 (b). To answer these questions, both intuitive analysis and empirical results are needed.
>
> The reviewer asks about our choice to use interaction tokens and nodes as bottlenecks for mixing the modalities. Our intuition for applying interaction through these bottlenecks was to allow both modalities to propagate individually, but then mix information between each other in a more controlled manner. A joint attention over all atoms (i.e., tokens, nodes) in both modalities would be more challenging to learn with limited fine-tuning datasets and be more computationally intensive. Empirically, In Table 7, we show that full connectivity on the graph side would yield lower performance than using the interaction tokens. Overall, we demonstrated the benefits empirically over prior methods such as QA-GNN, MHGRN, and RelNet.
>
> ERNIE is another powerful method for integrating KG information into LMs, but only integrates linked entity representations to the LM. Consequently, ERNIE does not retain explicit information from the graph structure, which our method can use for more structured reasoning across GNN layers (as suggested by the results in Table 5). We have added this distinction to the related work section.
>
> > There are also some related but missing references such as GLM[1], JAKET[2], CoLAKE[3]. Although they focus on language model pre-training, the idea of combining graph reasoning and language modeling is related.
>
> We thank the reviewer for pointing us to these works. We had cited them initially in their own Section of the related work, but removed them for space reasons because (as the reviewer points out), they were originally designed as pretraining methods. We’ve re-added a discussion on these methods in Section 2.
>
>
>
> > The technical contribution is somewhat small: except the multi-layer fusion part, others are very similar to QA-GNN including the GNN model.
>
> As language and reasoning abilities are closely intertwined (Mercier and Sperber, 2017), designing architectures that can adequately mix the representations of both modalities is a promising direction for designing improved reasoning systems for question answering. Our interactive scheme allows the bidirectional transfer of information from each modality to the other while retaining separate representation schemes for each modality. In this way, the method is applicable to any language model and any GNN architecture, which is a strength we demonstrate with new results in Appendix B, where switching out the language model for the same task still yields improved results.
>
> **References**
>
> [Mercier and Sperber, 2017] Mercier, H., & Sperber, D. (2017). The enigma of reason. Harvard University Press. https://doi.org/10.4159/9780674977860

---

> > ### Comment · Reviewer_RNQN · 2021-11-25
> > **Thanks for your response**
> >
> > Dear authors,
> >
> > Thanks for your response to my questions. However I still have the following concerns:
> >
> > > A joint attention over all atoms (i.e., tokens, nodes) in both modalities would be more challenging to learn with limited fine-tuning datasets > and be more computationally intensive. Empirically, In Table 7, we show that full connectivity on the graph side would yield lower
> > > performance than using the interaction tokens.
> >
> > What method in Table 7 do you refer to "full connectivity on the graph side"? If I understand correctly, "full connectivity on the graph side" is still different with "A joint attention over all atoms" over interaction layer. Do you have empirical results of the joint attention version of your model?
> >
> > > We’ve added these results to the paper in Table 6 using M= 4,5,6,7
> >
> > I suggest adding a figure showing the results of M from 1 to 7 to better demonstrate the effectiveness of number of interaction layer. Moreover, I'm wondering whether M=5 is the best for all the datasets or just Commonsense QA.

---

> > > ### Author Response · Authors · 2021-12-02
> > > **Additional comments**
> > >
> > > We thank the reviewer for continuing to engage with our paper!
> > >
> > > > What method in Table 7 do you refer to "full connectivity on the graph side"? If I understand correctly, "full connectivity on the graph side" is still different with "A joint attention over all atoms" over interaction layer. Do you have empirical results of the joint attention version of your model?
> > >
> > > Yes, that is correct -- we did not intend to make it seem as though these were the same modifications. We do not currently have results for the fully connected bi-directional joint attention between modalities. Our current results demonstrate that full connectivity on the graph side (i.e., all nodes in the subgraph, up to 200, are connected directly to the interaction mechanism, rather than merely nodes linked from the context) does not improve performance, illustrating that increased connectivity is not necessarily helpful in fine-tuning settings where there is limited data to learn the alignment between modalities. However, exploring fuller connectivity in a high-resource pretraining setting (as some of the references the reviewer mentioned do) continues to be an interesting direction for future work.
> > >
> > > > I suggest adding a figure showing the results of M from 1 to 7 to better demonstrate the effectiveness of number of interaction layer. Moreover, I'm wondering whether M=5 is the best for all the datasets or just Commonsense QA.
> > >
> > > We can run these additional layer numbers in the next few weeks and add them to the paper. We anticipate that *M* would be a dataset-specific hyperparameter that would need to be set depending on the typical complexity of question types in the dataset.
> > >
> > > Thanks again for your continued interaction!

---

### Official Review · Reviewer_99q1 · 2021-11-03

**Correctness:** 3
**Technical Novelty And Significance:** 3
**Empirical Novelty And Significance:** 3
**Recommendation:** 6
**Confidence:** 3

**Main Review:**

Strength:
- A novel architecture to enable deeper interaction between LM and GCN.
- Clear explanation of the proposed model.
- Extensive ablation studies to demonstrate the importance of modality interaction layer
- selection, parameter sharing, graph connectivity, and parameter initialization.
-  Consistent performance improvement over the baseline models on all of the three QA benchmarks.

Weakness:
- paper title is misleading, not directly related to LM
- no citation and description for the baseline method T5+KB (Table 4)
- As shown in Table 7, the proposed method is very sensitive to many factors.  The performance drops significantly when we change any of them.
- The data preprocessing and training steps are complex.


Questions:
- According to the parameters presented in Table 8, the knowledge from LM and GNN are only fused at the last 5 layers (parameter M) when 24-layer LMs are used, and at the last 3 layers when 12-layer LMs are used. It may be useful to show how the performance changes when using different M.
- According to [2], the CommonsenseQA IH-dev set contains 1,221 questions in total. May I know many questions are in each data split shown in Table 5?
- Is there any case that no entities are found during KG retrieval (Sec. 3.1)? If there are any, how do you handle these cases? What’s the percentage of such cases.

Reference:
[1] Michihiro Yasunaga, Hongyu Ren, Antoine Bosselut, Percy Liang, and Jure Leskovec. QA- GNN: Reasoning with language models and knowledge graphs for question answering. ArXiv, abs/2104.06378, 2021.
[2] Bill Yuchen Lin, Xinyue Chen, Jamin Chen, and Xiang Ren. Kagnet: Knowledge-aware graph networks for commonsense reasoning. In Empirical Methods in Natural Language Processing (EMNLP), 2019.


**Summary Of The Paper:**

In this paper, the authors introduce a new model for the QA tasks. They improve the existing approaches by adding the interaction components (the interaction token for the LM and interaction node for the GNN) to fuse knowledge from LM and knowledge graph in a more interactive manner. The LM can be any Transformer encoder-based pretrained LMs, where they tested RoBERTa-Large, AristoRoBERTa, and SapBERT in the experiments. The GNN is a simplification of the method of [1]. Experiments have been conducted to compare the proposed model and the prior KG+LM and LM-only baselines. Their model proved to be effective on three benchmarks, CommonsenseQA, OpenbookQA and MedQA-USMLE.


**Summary Of The Review:**

In summary, the paper has some nice contributions, such as a novel architecture to enable deeper interaction between LM and GCN, clear writing, and extensive analysis. Although there remain some issues and questions, the paper should be able to bring to the community some new aspects.

---

> ### Author Response · Authors · 2021-11-19
> **Response to Reviewer 99q1**
>
> We thank the reviewer for describing our work as novel, rigorous, and clearly presented. We address some the reviewers points in the general response above, and the remainder below:
>
> > As shown in Table 7, the proposed method is very sensitive to many factors. The performance drops significantly when we change any of them.
>
> The reviewer mentions that our ablation study in Table 7 shows decreased performance when these dimensions of our model are changed. However, we chose to depict an ablation study in Table 7 that was designed to test critical portions of the model, and show which components are necessary. As the modality interaction is a major contribution of our work, we are in fact satisfied to see a significant performance drop when it is removed. The other ablations -- removing parameter sharing, increasing the connectivity of the graph, and initializing the nodes randomly -- would drastically increase the amount of parameters in the model learned from scratch, which could be expected to yield lower results given the limited amount of training data in these benchmarks.
>
> Following the suggestion of the reviewer, we’ve also added the number of GreaseLM layers, M, as an additional ablation in this table. As can be seen, the model is already less sensitive to this hyperparameter.
>
> > The data preprocessing and training steps are complex.
>
> Our data preprocessing pipeline follows prior published work (MHGRN, Feng et al., 2020; QA-GNN, Yasunaga et al., 2021).
>
> Our training objective is the cross-entropy objective typically used for training on CommonsenseQA (and most MCQA datasets), and our model is trained end-to-end using only this objective. There are no intermediate or multi-task objectives.
>
> > no citation and description for the baseline method T5+KB (Table 4)
>
> There is no published (or preprint) reference for this method. However, we’ve added a link to the page where the method is described: https://leaderboard.allenai.org/open_book_qa/submission/brhieieqaupc4cnddfg0
>
> > Ablation of number of GreaseLM layers
>
> We’ve added these results to the paper using M= 4,5,6,7 and find that performance is pretty stable across M = 4,5,6 with a slight best performance on M=5. Performance begins to drop when M increases to 7.
>
> > Why does performance go up as number of prepositional phrases increases
>
> We’ve addressed this with a Table in general response (and added this content to the paper).
>
> > Is there any case that no entities are found during KG retrieval (Sec. 3.1)? If there are any, how do you handle these cases? What’s the percentage of such cases.
>
> For CommonsenseQA and OpenbookQA, there are no cases where no entities are found during KG retrieval. For MedQA, 5% of cases do not yield a retrieved entity. In these cases, we represent the graph using a dummy node initialized with 0. In essence, GreaseLM backs off to only using LM representations as the graph propagates no information. We’ve added this information to the appendix.
>
> **References**
>
> [Feng et al., 2020] Feng, Y., Chen, X., Lin, B. Y., Wang, P., Yan, J., & Ren, X. (2020). Scalable multi-hop relational reasoning for knowledge-aware question answering. EMNLP 2020.
>
> [Yasunaga et al., 2021] Yasunaga, M., Ren, H., Bosselut, A., Liang, P., & Leskovec, J. (2021). QA-GNN: Reasoning with Language Models and Knowledge Graphs for Question Answering. NAACL 2021.

---

> > ### Comment · Reviewer_99q1 · 2021-12-01
> > **Thanks for your response.**
> >
> > I have read the author's response, and some of my concerns are not solved. Therefore, I keep my score.

---

> > > ### Author Response · Authors · 2021-12-02
> > > **Thanks for engaging!**
> > >
> > > We thank the reviewer for reading our response. In response to the weaknesses laid out by the reviewer in their original review, we:
> > >
> > > * Changed the title of the paper to reflect the focus on QA
> > > * Added a reference to the source of the T5+KB model
> > > * Explained that Table 7 is an ablation study, so we’d expect that performance would drop when these components are changed
> > > * Clarified that our data preprocessing pipeline follows prior work (Feng et al., EMNLP 2020; Yasunaga et al., NAACL 2021) and our training consists of a typical MCQA training objective.
> > > * Added additional content in Tables 5 and 6 to address the reviewer's questions about the effect of the number of layers and prepositional phrases.
> > >
> > > We hope these changes addressed some of the weaknesses outlined by the reviewer.

---

### Official Review · Reviewer_ZyPu · 2021-11-03

**Correctness:** 4
**Technical Novelty And Significance:** 3
**Empirical Novelty And Significance:** 3
**Recommendation:** 8
**Confidence:** 4

**Main Review:**

Overall I like this paper since the method is simple and the results are good. I have only one problem: the initialization of LMs. The authors say that they initialize different LMs for different datasets to show that the method is agnostic to them. I am not convinced by this argument. A better way to do this would be to experiment with different LMs on one dataset to show that the method is agnostic to it and then use the best one for the other dataset(s) (Note: I understand that this would not apply to the biomedical dataset).

Other than this, I enjoyed reading the paper and liked the analysis and the detailed ablations. I am willing to increase my rating if the above concern is resolved.

EDIT: changed my rating after the author response.

**Summary Of The Paper:**

This paper presents a KG augmented LM which fuses graph and contextual representations. The model, when evaluated on multi-choice QA, outperforms strong baselines.

The context, question, answer pairs are encoded with a standard LM and the nodes of a relevant subgraph which is retrieved from a KG are encoded with a GNN. The representations of (i) a special interaction token (same as CLS), and (ii) a special interaction node, are input to custom layers that mix/fuse them which potentially makes them more informative to the downstream MLP head.

The authors experiment on three MCQA datasets: commonsenseQA, openbookQA, and medQA-USMLE (from the biomedical domain). ConceptNet is used as the KG for the first two datasets and a self-constructed KG is used for the third. The evaluation and analysis show that this method outperforms strong baselines (sometimes with a much higher parameter count). especially when the questions are complex and have prepositional phrases, negation terms, etc.

**Summary Of The Review:**

A good paper with a small and easily fixable issue regarding LM initialization.

---

> ### Author Response · Authors · 2021-11-19
> **Response to Reviewer ZyPu**
>
> We thank the reviewer for their kind words about the quality of our paper. We address their point about LM initialization below.
>
> > Running a different LM on the same dataset
>
> The reviewer points out that an interesting experiment would be to test whether GreaseLM improves results regardless of the LM initialization for the SAME task, rather than using a different LM for different tasks. Following this suggestion, we ran experiments on RoBERTa-base for CommonsenseQA and BioBERT and PubmedBERT for MedQA (in addition to the initial experiments). Across all LMs, we find that GreaseLM provides similar improvements as for the original language models tested in these settings. We’ve added these results to the Appendix of the paper.
>
> **CommonsenseQA**
>
> | Model           |  IH-Test Acc.        |
> |-------------------------|--------------|
> | RoBERTa-base            | 59.8         |
> | RoBERTa-base + GreaseLM | **65.0**         |
>
> **MedQA-USMLE**
>
>
> | Model                   | Test Acc.    |
> |-------------------------|--------------|
> | BioBERT                 | 34.1         |
> | BioBERT + GreaseLM      | **34.6**         |
> ||
> | PubmedBERT              | 38.0         |
> | PubmedBERT + GreaseLM   | **38.7**         |

---

> > ### Comment · Reviewer_ZyPu · 2021-11-25
> > **Change in rating**
> >
> > I have taken into consideration the authors' response and changed my rating.

---

> > > ### Author Response · Authors · 2021-12-02
> > > **Thank you for the discussion**
> > >
> > > We thank the reviewer for considering our rebuttal and updating their score.

---

### Official Review · Reviewer_NQbJ · 2021-11-05

**Correctness:** 4
**Technical Novelty And Significance:** 3
**Empirical Novelty And Significance:** 3
**Recommendation:** 8
**Confidence:** 3

**Main Review:**

Strengths of the paper:

1. The paper presents a novel way of combining information from text and a KB in a bidirectional way.
2. The results presented in the paper show strong gains against baseline methods on 3 different datasets.
3. Ablation studies show that the model achieves good performance on more complex questions.

Weaknesses, suggested improvements and requested clarifications

1. Regarding preposition phrases as a proxy for complexity: since the hypothesis is that the more the number of prepositional phrases in a question, the harder it is to answer. But from Table 5., the trend of performance seems to increase with the increased prepositional phrases (with 84.7 being the max for 4 PPs). It would be good if the authors could provide some discussion around this observation. Additionally, it would be good if the authors could report the number of examples (either raw number or as a fraction of the total dev set) for each of the categories: having that would help draw better conclusions.

2. The paper mentions that the entity extraction was done following Yasunaga et al. Was relevance scoring was done for the entities ?

3. Alongside with qualitative analysis, some quantitative analysis would be good to show what the model learns. Specifically, having the BFS analysis of the attention weights as a function of different GreaseLM layers (as done by Yasunaga et al.) can help demonstrate how having the bidirectional context information flow helps improve reasoning. Additionally, showing this for negations and / or examples which GreaseLM gets correct but QA-GNN does not (and vice-versa) can shed some light on what the model improves on (and what are the limitations).

4. Having an ablation on the number of GreaseLM layers would also be quite useful to answer if performance improves with more GreaseLM layers, are there diminishing returns or do we need just a few GreaseLM layers, beyond which it is detrimental to the model's performance.

5. The Graph connectivity ablation states that connecting the e_int node to all entities (instead of just the input text entities) hurts performance. It would be good if the authors could provide some intuition / insight as to why that might be the case.

Presentation Suggestions:

1. Page 5, line 1: should \tilde{e}^{(l-1)} be \tilde{e}^{l} instead ?

2. Page 5, Equations (6, 7, 8): should e^{l}_{s} and e^{l}_{j} be e^{(l-1)}_{s} and e^{(l-1)}_{j} respectively ?

Note to the authors:

There is a contemporaneous work submitted to ICLR 2022 [GNN is a Counter? Revisiting GNN for Question Answering](https://openreview.net/forum?id=hzmQ4wOnSb), whose hypothesis seems to be that by only using embeddings for node types and relation types, the models are able to attain good performance (86.67 acc on OpenBookQA) without needing any cross-modal information. While I understand this is contemporaneous work, but since the work is so relevant to this paper and seems to directly contradict the premise of this paper, it might be good to have a short discussion on this (just a suggestion).


**Summary Of The Paper:**

The paper presents a novel method of fusion of information from two modalities: text (context and question) and a Knowledge Base, for the task of question answering. Concretely, they propose GreaseLM layers that combine information from a Language Model (LM) and a Graph Neural Network (GNN) by mixing representations of embeddings from two special tokens (one in the LM and one in the GNN). The embedding in the LM passes information to the text modality via self attention; while the embedding in the GNN passes information to the KB modality via connections to other nodes. The modality interaction layer (dubbed MInt) is a simple MLP for combining the embeddings.
The paper demonstrates superior performance compared to baseline methods on 3 different datasets. Additional ablations quantitatively show that the model performs better on questions with negations, hedging and constraints (modelled with a proxy of number of prepositional phrases).

**Summary Of The Review:**

The paper presents a novel way of combining information from text and knowledge base modalities. The results on 3 datasets show the models improved performance, and ablations show quantitatively that the model performs better on complex questions. While I believe the paper can benefit from some qualitative analysis, and there are some open questions that need clarification, I believe the contributions presented are enough to merit an acceptance.

Edit: I have updated my score after reading the author response and edits made to the paper.

---

> ### Author Response · Authors · 2021-11-19
> **Response to Reviewer NQbJ**
>
> We thank the reviewer for their assessment on the novelty of our method and the recommendation for acceptance. We address their questions and concerns in the general response above and in the following remarks.
>
> > Why does connecting e_int to all nodes rather than just linked nodes hurt performance?
>
> The reviewer asks about the importance of the connectivity of the interaction node. We’ve added discussion on this point to the paper. Our hypothesis is that because the extracted subgraphs often contain up to 200 nodes, directly connecting all of them to the interaction node would overload the aggregate function that attends to all neighbors of the node. By connecting the interaction node only to linked entities, each linked entity serves as a filter for relevant information that reaches the interaction node.
>
> > The paper mentions that the entity extraction was done following Yasunaga et al. Was relevance scoring was done for the entities ?
>
> We’ve added more description of the entity retrieval in the appendix. We use relevance scoring as in Yasunaga et al., 2021 to prune the set of entities extracted as part of the subgraph (our maximum subgraph size is 200 nodes). However, we no longer use these node scores as part of the GNN message passing function.
>
> > Qualitative Analysis for what the model learns?
>
> The reviewer asks whether qualitative analysis supports our hypotheses as well. We will add an example in the paper, but we observe cases where the change in attention weights by GreaseLM over the course of multiple layers of propagation reflects a correct change in focus to reason about the situation. For example, in the context:
>
> What is unlikely to get *bugs* on its *windshield* due to *bugs*’ inability to reach it when it is moving?
>
> **(a) airplane** (b) scooter (c) motorboat (d) car (e) motor vehicle
>
> We observe that the attention weight by the interaction node increases on the “bug” entity in the intermediate GreaseLM layers, but drops again by the final one, resembling a suitable intuition of the hedge term “unlikely”. Meanwhile, the attention on “windshield” consistently increases across all layers. For QA-GNN, the attention weight for “bug” increases over the course of multiple layers (and “motor vehicle” is the final predicted answer). As “bug” is mentioned multiple times in the QA context, it may be well-represented in QA-GNN’s context node initialization, which is never reformulated by language representations, unlike in GreaseLM.
>
> > Discussion on “GNN is a Counter? Revisiting GNN for Question Answering”
>
> We thank the reviewer for pointing us to this contemporaneous work. While we’re also impressed by their empirical results, we’re surprised by the idea that node embeddings do not matter for representing commonsense knowledge graphs. This hypothesis would imply that graph structure is the crucial component for reasoning over commonsense KGs, which is counterintuitive given their underspecification and sparsity (Malaviya et al., 2020). Additionally, the authors of this work achieve the best results using only graphs containing linked entities to the context (i.e., no extracted neighborhoods), which seems to discard most of the aforementioned structural augmentation too. We’d be interested in seeing whether the authors’ method extends to other datasets and domains and would have liked to have seen them release their code with their submission.
>
> > Ablation of number of GreaseLM layers
>
> We’ve added these results to the paper using M= 4,5,6,7 and find that performance is pretty stable across M = 4,5,6 with a slight best performance on M=5. Performance begins to drop when M increases to 7.
>
> > Why does performance go up as number of prepositional phrases increases ?
>
> We’ve addressed this with a Table in general response (and added this content to the paper).
>
> [Malaviya et al., 2020] Malaviya, C., Bhagavatula, C., Bosselut, A., & Choi, Y. (2020). Commonsense knowledge base completion with structural and semantic context. AAAI 2020.

---

> > ### Comment · Reviewer_NQbJ · 2021-11-28
> > **Update to scores**
> >
> > After reading the rebuttal presented, I am satisfied with the response and am updating my score. While there exists contemporaneous work that challenges the premise of this paper; as the rebuttal mentions, additional analysis on more datasets needs to be done.
> >
> > The qualitative analysis presented in the rebuttal is quite interesting, and I hope it makes its way to the final paper, since I believe it does add value to the current paper.

---

> > > ### Author Response · Authors · 2021-12-02
> > > **Thank you**
> > >
> > > We thank the reviewer for engaging with our rebuttal and updating their score. We agree that the qualitative analysis would strengthen the paper and will add it once we figure out what to remove given the page limit.

---

### Author Response · Authors · 2021-11-19
**General Response to all Reviewers**

We thank the reviewers for their positive feedback on our paper and are excited to incorporate their recommendations and suggestions. Based on reviewer feedback, we’ve made the following changes to the paper: (highlighted in blue in the paper text, except for paper title):

Changed the title to reflect focus on QA
Added related work discussion around ERNIE, CoLAKE, JAKET, other methods that integrate KG information into LMs
Added number of examples per category to Table 5 to demonstrate the significance of performance differences
Added number of GreaseLM layers hyperparameter to Table 6 to show the model is fairly consistent across this hyperparameter
Added additional discussion on importance of connectivity to interaction node being limited to linked entities in Section 5.2
Added LM-agnosticity experiment to Appendix (Tables 9, 10), to demonstrate our method extends to different pretrained LMs for the same task
Added more description of entity linking to Appendix A to clarify how the initial KG is retrieved

We’ve also responded to reviewers individually, with certain comments being addressed jointly below (but also repeated in individual reviewer comments).

> Ablation of number of GreaseLM layers (NQbJ, 99q1)

The reviewers asked how many GreaseLM layers (M) were needed for best performance and how performance would vary across different values. We’ve added these results to the paper in Table 6 using M= 4,5,6,7 and find that performance is pretty stable across M = 4,5,6 with a slight best performance on M=5. Performance begins to drop when M increases to 7.

> Why does performance go up as number of prepositional phrases increases (NQbJ, 99q1)

Certain reviewers asked why performance went up for questions with more prepositional phrases if these questions were supposed to be more complex (i.e., more difficult). We’ve added the number of examples in each category for the development set in Table 5 and provide the numbers here for clarity.

|    | PP = 0 | PP = 1 | PP = 2 | PP = 3 | PP = 4 | Negations | Hedges |
|---|--------|--------|--------|--------|--------|-----------|--------|
| n |    210 |    429 |    316 |    171 |     59 |        83 |    167 |

We note that PP=1 and PP=2 are the most common examples in the training distribution too and may be the ones that are easiest to learn, yielding higher performance across models. Performance drops on PP=3. PP = 4 performance is higher, but the number of examples in this set is lower, making the higher numbers for each model potentially less reliable. Across all numbers of prepositional phrases except P=0, the performance of GreaseLM exceeds QA-GNN.

> Paper Title (99q1, RNQN)

Certain reviewers expressed concern about the generality of the paper title given the focus on QA. We thank the reviewers for their feedback on the title and have updated it to “GreaseLM: Graph Reasoning Enhanced Language Models For Question Answering” to better reflect the tasks we evaluate on.

---

### Decision · Program_Chairs · 2022-01-20

**Decision:**

Accept (Spotlight)

**Comment:**

The paper presents a novel method of fusion of information from two modalities: text (context and question) and a Knowledge Base, for the task of question answering. The proposed method looks quite simple and clear, while the results show strong gains against baseline methods on 3 different datasets. Ablation studies show that the model achieves good performance on more complex questions. While the reviewers raise some concerns, e.g., on the sensitivity of the proposed method, the technical novelty against prior works, they see values in this paper in general. And the authors did a good job in their rebuttal. After several rounds of interactions, some reviewers were convinced to raise their scores by a little bit. As a result, we think the paper is in a good shape and ICLR audience should be interested in it.